# Revisiting Resveratrol as an Osteoprotective Agent: Molecular Evidence from In Vivo and In Vitro Studies

**DOI:** 10.3390/biomedicines11051453

**Published:** 2023-05-16

**Authors:** Haryati Ahmad Hairi, Putri Ayu Jayusman, Ahmad Nazrun Shuid

**Affiliations:** 1Department of Biochemistry, Faculty of Medicine, Manipal University College Malaysia, Jalan Batu Hampar, Bukit Baru, Melaka 75150, Malaysia; haryati.hairi@manipal.edu.my; 2Department of Craniofacial Diagnostics and Biosciences, Faculty of Dentistry, Universiti Kebangsaan Malaysia, Kuala Lumpur 50300, Malaysia; putriayu@ukm.edu.my; 3Department of Pharmacology, Faculty of Medicine, Universiti Teknologi Mara (UITM), Jalan Hospital, Sungai Buloh 47000, Malaysia

**Keywords:** resveratrol, osteoporosis, SIRT1, oxidative stress, osteoblast, osteoclast

## Abstract

Resveratrol (RSV) (3,5,4′-trihydroxystilbene) is a stilbene found in abundance in berry fruits, peanuts, and some medicinal plants. It has a diverse range of pharmacological activities, underlining the significance of illness prevention and health promotion. The purpose of this review was to delve deeper into RSV’s bone-protective properties as well as its molecular mechanisms. Several in vivo studies have found the bone-protective effects of RSV in postmenopausal, senile, and disuse osteoporosis rat models. RSV has been shown to inhibit NF-κB and RANKL-mediated osteoclastogenesis, oxidative stress, and inflammation while increasing osteogenesis and boosting differentiation of mesenchymal stem cells to osteoblasts. Wnt/β-catenin, MAPKs/JNK/ERK, PI3K/AKT, FoxOs, microRNAs, and BMP2 are among the possible kinases and proteins involved in the underlying mechanisms. RSV has also been shown to be the most potent SIRT1 activator to cause stimulatory effects on osteoblasts and inhibitory effects on osteoclasts. RSV may, thus, represent a novel therapeutic strategy for increasing bone growth and reducing bone loss in the elderly and postmenopausal population.

## 1. Introduction

Osteoporosis is a severe public health problem described by low bone mineral density (BMD T-score: −2.5 and lower) and bone mass, bone tissue degradation, and disturbance of bone microarchitecture, all of which increase the risk of fractures [1]. Increased bone resorption and lower bone formation rates induce the progression of bone loss. [2]. It is associated with high morbidity and mortality in which hormone deficiency [3], age-related disorders [4], steroid therapy [5], nicotine addiction [6], rheumatoid arthritis [7], and metabolic disorders (diabetes, dyslipidaemia, and metabolic syndrome) [8] can increase the risk of developing osteoporosis. Pharmacologic options for osteoporosis include antiresorptive agents such as bisphosphonates and denosumab, selective oestrogen receptor modulators (SERM) such as raloxifene and anabolic agents such as parathyroid hormone analogues, teriparatide, and abaloparatide [9]. However, the use of these pharmacologic agents is associated with rare serious side effects. For example, long-term use of bisphosphonate has been associated with an increased risk of atypical fracture and osteonecrosis of the jaw [10,11]. Apart from that, patients may also experience common adverse effects which are mainly related to gastrointestinal, cardiovascular, and endocrine systems [12]. Though bisphosphonate remains the first-line therapy for the treatment of osteoporosis, the concerns over its side effects limit patients’ compliance and trust which contributed to the decline in the use of this class of drug [10,11].

Resveratrol (RSV) (3,5,4′-trihydroxystilbene) is a natural polyphenolic stilbene compound found in the skin of red grapes, mulberries, peanuts, pines, and weed root extracts of *Polygonum cuspidatum* [13]. There is a wealth of in vitro, in vivo, and clinical evidence on the pharmacological properties of RSV such as antioxidant [13], anti-inflammatory and antiapoptotic [14], antimicrobial [15], anticancer [16], phytoestrogenic [17], neuroprotective [18], cardioprotective [19], skin protective [20], and osteoprotective [21]. RSV could target intracellular receptors [22], enzymes [23], signalling molecules [24], antioxidant enzymes [25], and transcription factors [26]. Noteworthy, RSV is also able to regulate transduction signalling pathways such as the silent information regulator of transcription1 (SIRT1) and nuclear factor kappa-light-chain-enhancer of activated B cells (NF-κB) [27,28]. 

Some of RSV’s medicinal properties are directly related to its bone osteoprotective effects, which are further discussed below. RSV, which has potent antioxidant and anti-inflammatory properties, can mitigate the deleterious impacts of osteoporosis-induced imbalances in osteoblastogenesis and osteoclastogenesis. This review article focuses on the in vivo and in vitro experimental evidence of RSV efficacy in ameliorating bone loss and enhancing bone formation. This review also includes information on RSV’s molecular mechanism of action and its therapeutic potential as an osteoprotective agent. 

## 2. RSV Effects on Bone: In Vivo Evidence

It is desirable for RSV to enhance osteoblast-mediated bone growth while inhibiting osteoclast-mediated bone resorption for pharmaceutical value because it can act as both an anabolic and antiresorptive agent. RSV’s osteoprotective effects have been widely studied in animal models of osteopenia or osteoporosis, including ovariectomy, senescent, hindlimb suspension, glucocorticoid, and chemotherapy models. Generally, study designs are conceptually similar across the board, but the methods may differ significantly. For example, in the studies using the ovariectomy model, the route of administration of RSV may vary from oral feeding and oral gavage to subcutaneous injection.

Rats were ovariectomised bilaterally to simulate the model for postmenopausal osteoporosis. As in menopause, oestrogen deficiency after ovariectomy-induced rapid bone loss by accelerating bone resorption rate [29]. The senescent rat models represent senile osteoporosis, in which ageing results in gradual bone loss due to a decrease in bone formation rate [30,31]. Meanwhile, for the disuse osteoporosis model, hindlimb suspension induced similar bone changes to prolonged bed rest or physical inactivity in elderly humans, favouring adipocyte differentiation over osteoblast differentiation [32]. In these osteoporosis models, RSV has exhibited bone-conserving effects, which is intriguing.

There were 10 published in vivo studies utilising an ovariectomized (OVX) rat model to determine the osteoprotective effects of RSV on postmenopausal osteoporosis. In these studies, the RSV dose varied between 0.625 µg/kg/day to 500 mg/kg/day. According to Zhang et al. [33], an oral dose of 5 mg/kg/day of RSV for 8 weeks only managed to increase the serum calcium and phosphorus levels. There were no significant changes in the other parameters such as BMD, alkaline phosphatase (ALP), amino-terminal propeptide of type 1 procollagen (PINP), and oestrogen. RSV begins to demonstrate osteoprotective effects at doses of 15 and 45 mg/kg/day. At these doses, the bone formation markers, BMD and PINP, were significantly increased, while bone resorption markers such as tartrate-resistant acid phosphatase (TRAP) and C-terminal telopeptides type I collagen (CTX-1) were significantly decreased compared to the OVX control group. The study results also indicated that RSV increased oestrogen serum levels, which may be responsible for the inhibition of osteoclastogenesis and improved BMD [33].

Postmenopausal osteoporosis is associated with trabecular disproportionate loss rather than cortical bone [34]. According to Zhao et al. [35], oral RSV doses of 40 and 80 mg/kg/day orally for 12 weeks were able to increase the BMD of the neck and distal end of the femur. The trabecular microarchitecture of the right distal femur improved, with a higher percentage of the trabecular area (Tb.Ar), trabecular thickness (Tb.Th), trabecular number (Tb.N), and lower trabecular separation (Tb.Sp), indicating that RSV increased bone density. Furthermore, the RSV dose of 80 mg/kg/day did not cause any significant stimulation of the uterus. RSV treatments were also found to reduce bone resorption and proinflammation markers such as TRAP, interleukin-6 (IL-6), and tumour necrosis factor alpha (TNFα) by inhibiting the receptor activator of nuclear kappa B ligand (NF-κB) pathway. It also downregulated the RANKL/OPG ratio. [35]. Surprisingly, even the smallest dose of RSV (0.625 g/kg/day) was able to reduce the RANKL/OPG ratio and proinflammatory cytokine levels [36].

In another study by Feng et al. (2018), subcutaneous administration of RSV at 40 mg/kg/day for 10 weeks increased the BMD and trabecular parameters of the OVX rats. This was probably achieved by decreasing oxidative stress, reactive oxidant species (ROS), and malondialdehyde (MDA) levels and increasing antioxidant status, superoxide dismutase (SOD), and glutathione (GSH) levels [37]. Jiang et al. [38] also discovered that intraperitoneal injection of RSV at the same dose managed to increase the expression of bone formation markers such as ALP, runt-related transcription factor 2 (Runx2), and osterix (Osx) via the SIRT1 pathway. These findings were corroborated by Feng et al. [39], which demonstrated that oral RSV promoted osteoblast differentiation via the SIRT1-NF-kB signalling pathway. 

SIRT1 and Wnt/-catenin are the signalling pathways which regulate Runx2 and PPARγ levels [40,41]. Elseweidy et al. [42] discovered that SIRT1 and Wnt/β-catenin expressions were reduced after ovariectomy but recovered with RSV treatment. This could explain why the OVX control group had higher levels of proliferator-activated receptor gamma (PPARγ) but lower levels of Runx2 than the RSV groups. These findings reflected that osteoporotic rats had higher adipogenic differentiation than osteogenic differentiation. RSV administration to these ovariectomised rats improved osteogenic differentiation and suppressed adipogenic differentiation, probably by increasing Runx2 and decreasing PPARγ levels via the SIRT1 and Wnt/-catenin signalling pathways [42]. 

The BMP-2/Smad/Runx2 signalling pathways, which govern bone production, are linked to osteoblast development and extracellular matrix synthesis [43]. However, the BMP-2/Smad/Runx2 signalling cascade activity was reduced in the bone tissues of OVX rats, exacerbating osteoporosis [44]. Suppression of miR-92b-3p expression increased NADPH oxidase 4 (Nox4) and NF-κB levels, decreased Smad7, BMP2, and Runx2 gene and protein expressions, and inhibited bone marrow mesenchymal stem cell (BMSC) proliferation and osteoblast differentiation, resulting in more severe osteoporosis. On the other hand, RSV elevated miR-92b-3p level, thus reducing the Nox4/NF-κB activity and increasing BMP-2/Smad/Runx2 activity [45]. In another finding, Guo et al. [46] reported that RSV suppressed miR-338-3p, leading to an increase in Runx2 expression in human osteoblast cells.

Autophagy is a physiological process that preserves cell homeostasis by eliminating damaged organelles and proteins [47]. Recent research suggests that autophagy is vital in maintaining the equilibrium of bone metabolism. Notably, autophagy is involved in mineralisation and bone homeostasis in osteoblasts; specifically, autophagic vacuoles have been reported to act as vehicles for mineralisation matrix secretion. In summary, autophagy modulation is a critical factor in osteoporosis [48,49]. Wang et al. [50] discovered that ovariectomy causes inhibition of autophagy in osteoblasts while activating it in osteoclasts. RSV may protect bone by reversing the ovariectomy-induced changes in autophagy in a dose-dependent manner.

Senile osteoporosis has a different pathophysiology than postmenopausal osteoporosis [51]. Evidence suggests that as we age, BMSCs differentiate more into adipocytes than osteoblasts and enter senescence, resulting in decreased bone formation and contributing to senile osteoporosis. As a result, bone marrow fat accumulation occurs at the expense of osteoblastogenesis. Thus, a low bone formation rate is one of the characteristic hallmark features of senile osteoporosis and affects mostly cortical bones [52]. Aged male rats were used as the senile osteoporosis model. Tresguerres et al. used 22-month-old rats [53] but Ameen et al. [54] and Lee et al. [55] used four to six months old rats. It is noteworthy that at the age of six to nine months, rats have a stable level of bone turnover. Beyond the age of nine months, rat bones are undergoing the ageing process [56]. Oral RSV, given to aged rats at the dose of 10 mg/kg/day for 10 weeks, was found to increase bone volume, trabecular, and cortical thickness. However, there were no changes in plasma CTX or osteocalcin level [53]. A report from Ameen et al. [54] demonstrated that oral RSV given to aged rats at the dose of 20 mg/kg/day for six weeks was able to significantly reverse the age-dependent osteoporotic changes seen in femur epiphysis, metaphysis, and diaphysis. This study found that RSV could counteract the molecular changes in aged male osteoporosis by decreasing inflammatory cytokines such as Il-1 β, Il-6, TNF-α, and RANKL, as well as oxidative stress markers such as MDA and nitric oxide (NO), and by increasing Forkhead box O1 (FoxO1), Sirt1, and OPG gene expressions [54]. However, Lee et al. [55] reported that six-month-old rats treated with RSV at a dose of 20 mg/kg/day for three months had no significant effect on bone volume and bone formation markers such as ALP, osterix, and osteocalcin. These findings suggested that a low dose of RSV has no effect on the male skeleton during the ageing process. The choice of the age of rats used for the senile osteoporosis model may also affect the study results.

The hindlimb suspension (HLS) rat model caused bone changes similar to those caused by prolonged bed rest in humans. Aside from that, the hindlimb unloaded rat model was used to investigate bone loss caused by mechanical unloading. Inactivity and bed rest reduce mechanical loading, which accelerates age-related bone loss in the elderly [57]. Durbin et al. [58] combined HLS and fed aged male rats with 12.5 mg/kg of RSV daily for one week prior to HLS and for the entire two weeks of HLS. RSV supplementation reduced mechanical unloading-induced femoral BMD, calcium, and phosphate loss. RSV supplementation also prevented microarchitectural deterioration induced by mechanical unloading [58].

Glucocorticoid-induced osteoporosis is the most common cause of secondary osteoporosis [59]. In the study by Yang et al. [60], glucocorticoid-induced osteoporosis can be induced in male rats by administering dexamethasone intramuscularly at a dose of 5 mg/kg twice weekly for six weeks. RSV at oral doses of 5 and 45 mg/kg increased BMD of the right femur while lowering femoral density, ALP, and osteocalcin levels. Western blot study revealed that high-dose RSV treatment resulted in downregulation of phospho-Akt (*p*-Akt) and phospho-mammalian target of rapamycin (*p*-mTOR) and upregulation of Sirt1, LC3, and Beclin-1 expressions in osteoblasts. These findings showed that RSV administration increased mitophagy by modulating SIRT1 and the PI3K/AKT/mTOR signalling pathways, protecting osteoblasts from the deleterious effects of glucocorticoid [60].

Irreversible bone deterioration is one of the most serious health hazards linked with chemotherapy treatments [61]. A study by Lee et al. [62] found that acute methotrexate (MTX) treatment resulted in significantly reduced growth plate thickness, shorter primary spongiosa height, reduced trabecular bone volume, and increased marrow adiposity. In this osteoporosis rat model study, RSV supplementation protected against MTX-induced bone damage by inhibiting osteoclastogenesis (reducing TRAP levels) and lowering proinflammatory cytokines such as TNF, IL1, and IL6. RSV also maintained the expression of osteogenesis genes and decreased expressions of adipogenic and osteoclastogenic factors [62].

Together, these in vivo investigations have demonstrated that RSV improves bone quality in animals with oestrogen deficiency-induced osteoporosis, senile osteoporosis, and secondary osteoporosis by increasing BMD, trabecular and cortical bone microstructure, bone strength, and bone histomorphometric parameters (summarised in Table 1). It has been proposed that RSV could regulate osteoblastogenesis by activating the expression of the SIRT1 protein. Given RSV’s potential in vivo skeletal benefits, we investigated its effects on bone cells as well as the molecular mechanisms underlying these effects.

### 2.1. The Effects of RSV on Osteoblast-Bone Forming Cells

RSV has been studied in vitro for its effects on cell viability, proliferation, mineralization, and the expression of osteogenic genes in the cell models including MC3T3-E1, human osteoblast cells (HOb), primary human, rat, mouse, and bovine osteoblast cells. In vitro studies showed that RSV can directly stimulate the proliferation and differentiation of osteoblasts. RSV stimulated the expression of osteoblast-specific genes such as osterix, Runx-2, type 1 collagen (COL-1), bone morphogenetic protein-2 (BMP-2), osteocalcin (OCN), and osteopontin (OPN), which are mainly responsible for regulating the proliferation and differentiation of osteoblasts. In parallel, RSV increased cell proliferation, alkaline phosphatase (ALP) activity, collagen synthesis, and calcium deposition in osteoblast cells in a dose-dependent manner. 

Bone marrow generates multipotent bone marrow mesenchymal stem cells (BMSCs). They have the ability to self-renew and differentiate into a range of cell types such as osteoblasts, chondroblasts, adipocytes, neuroblasts, and myoblasts. [63]. Using rat or human bone marrow-derived MSCs, various concentrations of RSV were demonstrated to promote proliferation, differentiation, and mineralisation of the cells into the osteoblastic lineage [64]. They also increased the expressions of ALP, osterix, Runx-2, COL1, BMP-2, OPN, OCN, and calcium deposition by the MSCs. Moreover, RSV-enhanced proliferation and osteoblast differentiation from MSCs via ER-dependent ERK 1/2 activation and Wnt/β catenin signalling pathway [65,66].

With successive cell divisions, BMSCs will reach the ‘Hayflick limit’ of replicative senescence. The biological functions of residual MSCs deteriorate over time, making them vulnerable to cellular damage and senescence [67]. Previous research has shown that senescent cells have higher levels of reactive oxygen species (ROS) than normal cells, implying that ROS plays a role in cell senescence [68]. Excessive ROS production in BMSCs caused oxidative stress, which suppressed osteogenic lineage and increased adipogenic terminal differentiation [69]. Intriguingly, RSV treatment at doses ranging from 5 to 25 g/mL, significantly reduced cell senescence as measured by senescence β-galactosidase staining and related genes, p16, p21, and p53 [70]. High-altitude hypoxia causes massive free radical and oxidative stress damage, which may induce osteoporosis [71]. According to a study by Yan et al. [72], a hypoxic microenvironment inhibited BMSC proliferation and significantly reduced calcium deposition and ALP activity. The expression of genes involved in osteoblastogenesis was also decreased significantly. Under a high-altitude hypoxic environment, RSV treatment could effectively promote BMSC proliferation and osteoblastogenesis [72]. In other studies, RSV reduced ROS level and protected MC3T3-E1 osteoblast cells from hydrogen peroxide (H_2_O_2_) and cadmium-induced oxidative stress [73,74]. 

In stem cell differentiation, the metabolic change from glycolysis to mitochondrial oxidative phosphorylation (OXPHOS) has recently acquired importance [75,76]. Glycolytic metabolism is the primary mode of self-renewal and maintenance in proliferating stem cells. To fulfil the increased energy (ATP) demand, stem cells switch from glycolytic to mitochondrial OXPHOS metabolism [77]. OXPHOS occurs within the mitochondria where the mitochondrial respiratory complexes and the ATP synthase create ATP. The catalytic core components of mitochondrial respiratory complexes, as well as the ATP synthase (mtDNA), are encoded by DNA. If cells require more ATP during stem cell differentiation, they must increase mitochondrial biogenesis, which results in increased mitochondrial mass and mtDNA content [78]. Moon et al. [79] discovered that RSV treatments increased mitochondrial mass and mtDNA content in periosteum MSCs (POMSCs) during osteogenic differentiation. Furthermore, RSV upregulation of mitochondrial biogenesis in POMSCs is consistent with the idea that a metabolic shift from glycolysis to mitochondrial OXPHOS occurs during stem cell differentiation [79]. Based on these findings, RSV, a small molecule that promotes mitochondrial biogenesis, could be a new modulator of adult POMSC osteogenesis in regenerative medicine.

Lipopolysaccharide (LPS), a significant component of the outer membrane of Gram-negative bacteria, has been used to analyse experimentally caused infection, inflammation, or tissue damage [80]. Previous research found that LPS significantly increased the production of inflammatory molecules, lipid mediators, and adhesion molecules such as nitric oxide (NO), prostaglandin E2 (PGE2), tumour necrosis factor-alpha (TNF-a), interleukin-1 (IL-1), ROS, inducible nitric oxide synthase (iNOS), and cyclooxygenase-2 (COX-2) [81,82]. Recent evidence has shown that LPS could cause diabetes [83], chronic obstructive pulmonary disease [84], neurodegenerative diseases [85], and osteoporosis [86] by activating numerous inflammatory cells. On the other hand, RSV increased the expression of osteoblast-specific genes such as ALP, Runx-2, OPN, and OCN by activating the SIRT 1 pathway, which reversed LPS-induced inhibition of osteoblast proliferation and differentiation [87]. The promotion of osteoblast differentiation by RSV also occurred independently of inflammation [88].

Bisphosphonates are major pharmacological agents that suppress osteoclast function, hence reducing bone resorption and loss. They are commonly used in clinical practice to treat skeletal disorders including osteoporosis, Paget disease of bone, and multiple myeloma [89]. Long-term bisphosphonate treatment has been linked to pathologic conditions such as osteonecrosis of the jaw (BRONJ), which impairs the bone regeneration process [90]. Borsani et al. [91] found that RSV reversed zoledronate-suppressed human osteoblast differentiation and its associated low BMP2 expression. Notably, BMP-2 is one of the most studied members of the BMP family and has been identified as the most effective inducer of osteogenesis [92]. 

Prostaglandins have been shown to act as autocrine/paracrine regulators of osteoblasts and play important roles in bone metabolism regulation [93]. Prostaglandin E2 (PGE2) and prostaglandin F2α (PGF2α) are well-known bone resorptive agents [94,95]. However, PGE2 and PGF2α are now recognised as bone remodelling mediators that regulate a wide range of intracellular signalling pathways in osteoblasts [95,96]. A previous study reported that PGE2 and PGF2α induced IL-6 secretion via activation of p44/p42 mitogen-activated protein (MAP) kinase and p38 MAP kinase in MC3T3-E1 cells [97]. According to Yamamoto et al. [98] and Kuroyanagi et al. [99], RSV markedly reduced the PGE2 and PGF2α-stimulated OPG synthesis through the inhibition of p44/p42 MAP kinase, p38 MAP kinase, and SAPK/JNK in osteoblasts. RSV is classified as a phytoestrogen since it causes varying degrees of oestrogen receptor agonism in various cells [100,101]. According to Shah et al. [102], RSV at 0.1 M boosted ER expression in a time-dependent manner, peaking after 48 h without altering ER isoform expression. RSV interacted with the catalytic amino acid triad of the ER pocket (Hie524, Phe404, and Glu353) with favourable binding energy of contact with ER isoforms. The manner of RSV binding was identical to that of the prototype 17β-estradiol [102]. Therefore, the oestrogen mimetic action of RSV indicated its therapeutic potential as a bone anabolic agent for the treatment of postmenopausal osteoporosis.

Other findings from Tseng et al. [103] demonstrated that RSV promoted osteogenesis of human MSCs by activating the master osteogenic transcription factor, RUNX2 via SIRT1 and its interactions with FoxO3A. However, RSV inhibited adipogenesis by deactivating the master adipogenic transcription factor, PPARγ. The binding of the SIRT1-FoxO3a complex to the novel FoxO response element (FRE) site on the RUNX2 promotor directly linked FoxO3a to osteogenesis [103]. Concomitantly, RSV suppressed TNFβ expression by activation of SIRT1 and inhibition of NF-κB signalling pathway [104].

Most studies reported the RSV osteoblastogenesis-activating effects. There are a few contradictory findings, which could be attributed to the different cell types and RSV doses used. It is critical to delve deeper into the mechanisms of action that underpin RSV’s osteogenic potential.

### 2.2. The Effects of RSV on Osteoclast-Bone Resorbing Cells

Numerous studies have found that RSV inhibits the formation of osteoclast-like cells (evidenced by TRAP-positive multinucleated cells) and bone resorption pit. It also suppressed bone resorption markers in RAW264.7 cells including cross-linked C-telopeptide of type I collagen (CTX-1), matrix metalloproteinase-9 (MMP 9), and cathepsin K (CtsK).

Bone cells are harmed by inflammation and oxidative stress. RSV activity has been investigated in cells treated with lipopolysaccharides (LPS), hydrogen peroxide (H_2_O_2_), and doxorubicin (a first-line chemotherapeutic agent), all of which are known for their cytotoxicity and induction of osteoclast fusion and activation. According to Zong et al. [105], RSV significantly reduced LPS-induced expressions of NO, PGE2, iNOS, COX-2, TNF-α, and IL-1β in RAW 264.7 cells. Furthermore, RSV inhibited proinflammatory mediators and cytokines production in response to LPS by activation of phosphatidylinositol-3-kinase (PI3K/AKT) and SIRT1 expression. RSV also inhibited phosphorylation of cyclic AMP-responsive element-binding protein (CREB) and mitogen-activated protein kinase (MAPK) activation [105]. In RAW 264.7 cells exposed to H_2_O_2_, RSV improved oxidative stress status by increasing SOD and GSH-PX levels, while decreasing ROS and MDA levels [37]. RSV was also found to increase the expression of the antioxidant genes SOD 1 and Nrf2 in RAW264.7 cells exposed to doxorubicin [106]. RSV was also able to reduce the mRNA expression of the osteoclast fusion marker, Oc-stamp, as well as the osteoclast differentiation markers RANK, TRAP, and CtsK by regulating FOXO1 transcriptional activity and inhibiting the PI3K/AKT [37] and activated T-cells cytoplasmic 1 (NFATc1) [106] signalling pathways, which are important in osteoclastogenesis (Summarised in Table 2).

## 3. The Underlying Mechanisms of Action of Resveratrol as a Osteoprotective Age

The deposition and removal of damaged bone tissues, as well as the synthesis of new ones, has emerged as a means of maintaining bone homoeostasis. Together, these processes demonstrate that osteoclastic and osteoblastic activities are regulated by a series of interactions with their respective signalling pathways and osteogenesis-related gene/protein expressions. The osteoprotective properties of RSV through the modulation of RANK/RANKL/OPG, SIRT1, Wnt/β-catenin, MAPKs/JNK/ERK, PI3K/AKT, microRNAs, and BMP2 are discussed in this section as the key mechanisms related with bone remodelling and development of osteoporosis.

### 3.1. Regulating the RANK/RANKL/OPG System

Bone metabolism is an intricate phenomenon that is regulated not only by osteoblastic and osteoclastic activity but also by the tightly coupled action of OPG and RANKL in bone metabolism regulation [107]. Osteoclasts are multinucleated bone-resorbing cells and they begin as monocyte progenitors before committing to the macrophage lineage. Osteoblasts, on the other hand, promote osteoclastogenesis by secreting macrophage colony-stimulating factors (M-CSF), which works directly on osteoclast precursors [108]. The discovery of a molecular system consisting of RANK, also known as tumour necrosis factor (TNF) receptor superfamily, member 11a, NF-κB activator (TNFRSF11A), RANKL, also known as TNF (ligand) superfamily, member 11 (TNFSF11), and OPG, also known as TNF receptor superfamily member 11B (TNFRSF11B), have been critical in elucidating several important processes regulating bone biology [109]. RANKL is found in a variety of cells including osteoblasts, osteocytes, preosteoblasts, periosteal cells, dendritic cells, and vascular cells. It acts as a RANK ligand on the surface of osteoclasts. RANKL binds to its receptor RANK, allowing osteoclast activation, survival, and differentiation, while inhibiting osteoclast apoptosis. Maturation of osteoclasts occurs when RANKL produced by osteoblasts activates RANK in the osteoclasts. RANKL expression in osteoblasts is increased by bone resorption-stimulating hormones and cytokines. Mature osteoclasts also express RANK and RANKL, which both aid in osteoclast survival and stimulate bone-resorbing activity. Therefore, RANKL promotes and increases bone loss and osteoclastogenesis, and mice lacking RANKL developed osteoporosis due to osteoclast deficiency. Mature osteoclasts adhere to the bone surface and promote bone resorption by secreting acid and lytic enzymes (CtsK and TRAP) [110,111,112].

OPG is expressed primarily by osteoblasts and bone marrow stromal cells [113]. Several studies have found that the ratio of serum RANKL to OPG levels in the bone microenvironment is a key regulator of osteoclast formation and is important in bone metabolism diseases. OPG inhibited RANK activation by RANKL, reducing osteoclastogenesis [112,113]. OPG expression and production were regulated by cytokines such as IL-6 and TNFα [114], steroid hormones (17β estradiol) [115], transforming growth factor (TGFβ) [116], and bone morphogenetic proteins (BMPs) [117]. Glucocorticoids, prostaglandin E2, fibroblastic growth factor, and parathyroid hormone (PTH) were all known to suppress the expression of OPG [118]. By inhibiting the RANKL-RANK interaction, OPG inhibits osteoclast formation and osteoclastic bone resorption and activates bone formation. As OPG has a 500-fold higher affinity for RANKL than RANK, it prevents RANKL from binding to RANK. It inhibits osteoclastogenesis and protects the bone from osteoclast-induced resorption. Reduced OPG levels promoted not only osteoclastogenesis and bone resorption but also vascular Ca deposition [119].

Bone mass can be increased by inhibiting RANKL–RANK signalling in bone and thus preventing osteoclastic bone resorption [120,121]. In contrast, OPG-deficient mice developed severe osteoporosis as the result of the increased maturation stage of osteoclastogenesis [122,123]. These findings suggested that the RANK/RANKL/OPG pathway could be a potential therapeutic target in the treatment of bone metabolism disorders. Several cytokines can greatly affect the RANK/RANKL ratio by stimulating immune cell RANKL expression. It is now evident that inflammatory cytokines linked to osteoclastic bone loss, such as TNFα, IL-1, IL-6, and M-CSF, promoted RANKL but reduced OPG production. This was achieved by activating the RANK receptor on osteoclast precursors, making them more vulnerable to RANKL [124]. TNFα, IL-6, and IL-1 have been linked to the formation of osteoclasts in osteoporosis patients and animal models of ovariectomy [125,126]. A recent study found that IL-1 facilitated TNF’s osteoclastogenic action by boosting stromal cell RANKL expression and directly driving osteoclast precursor differentiation [127]. TNF also has a particular trait among inflammatory cytokines in that it not only boosted RANKL synthesis but also synergised with RANKL to accelerate osteoclastogenesis [128]. These effects are most likely attributable to the fact that RANKL belongs to the TNF superfamily and operates through the same pathways as TNF.

The effects of RSV on OPG and RANKL expressions have been widely documented. In the presence of RSV, the RANKL/OPG ratio decreased [35,36,37,42,54]. RSV increased bone mineral density in ovariectomised rats, which promoted osteogenesis by downregulating pro-inflammatory cytokines such as TNF, IL-1, IL-6, IL-23, IL-17a, IL1 and RANKL/OPG ratio [35,36]. In the 3-D osteogenic differentiation on collagen scaffold of rat adipose stem cells treated with RSV, an increase in OPG was observed along with an increase in the mineralisation process [64]. In addition, RSV inhibited the expression of osteoclast markers (RANK, NFATc1, TRAP, CtsK, and Oc-Stamp) in RAW264.7 cells to suppress the generation of osteoclast-like cells [106]. In an in vivo study, RSV reduced the osteoclastic differentiation marker (TRAP and CTX-1) by inhibition of NF-κB and NFATc1 signalling pathway [45]. The mechanisms underlying RSV’s action in orchestrating osteoclastogenesis and bone resorption through the RANK/RANKL/OPG system have been identified (Figure 1).

### 3.2. Regulating the SIRT1 and Wnt/β-Catenin Signalling 

Sirtuins are NAD+-dependent nicotinamide adenine dinucleotide (NAD+) deacetylases [129]. SIRT1 is the sirtuin protein with the largest molecular mass and the most thoroughly studied, capable of responding to a variety of biological activities such as regulating apoptosis, DNA damage repair, anti-inflammation, senescence, autophagy, and metabolism regulation in response to cellular energy and redox status. Intriguingly, SIRT1 can transduce metabolic signals into epigenetic changes by deacetylating histone and non-histone proteins which silences target proteins such as p53, FoxOs, and β-catenin at the transcriptional and post-translational levels [130,131]. 

SIRT1 expression is linked to bone mineral density and bone fragility. Previous research found that SIRT1 knockout mice have low bone formation capacity, demonstrating SIRT1′s importance in new bone formation [132]. Furthermore, SIRT1 knockout mice had lower bone mass and osteoblast-to-osteoclast ratio imbalance [133]. A clinical investigation found that the expression of SIRT1 was considerably reduced in the femoral necks of osteoporosis patients [134]. SIRT1 activity in peripheral blood mononuclear cells was reduced concurrently, indicating a close relationship between SIRT1 and osteoporosis [135]. Ageing is a risk factor for osteoporosis, and SIRT1 expression declines with age. Ma et al. [136] discovered a link between bone cell senescence and decreased SIRT1 expression. SIRT1 could, thus, be used as a biomarker for systemic bone metabolism and bone diseases.

RSV is a SIRT-1 activator. Several studies demonstrated that RSV treatment significantly increased the expression of SIRT-1 in osteoporosis rat model [38,39,42,60]. Similarly, Ma et al. [87] demonstrated that RSV stimulated osteoblast proliferation and differentiation via SIRT-1 activation. Furthermore, co-treatment with concentrated growth factor (CGF) and RSV significantly increased SIRT-1 levels in osteoblasts incubated with bisphosphonates, indicating both CGF and RSV may play a protective role in human osteoblasts [91].

Many studies have shown that SIRT1 is a promising bone homeostasis regulator gene that is crucial for bone marrow mesenchymal stem cells (MSCs) differentiation and viability of bone-forming cells and bone vasculature. It controls MSCs differentiation into osteoblasts by deacetylating several transcription factors including FoxO3a, β-catenin, and RUNX2. SIRT1 has been shown to promote osteoblast differentiation of MSCs by either directly deacetylating RUNX2 or indirectly modulating RUNX2 by forming a SIRT1-FoxO 3a complex and repressing PPARγ [137]. SIRT1-FoxO3a complex promoted osteogenic differentiation of MSCs over adipogenesis [138]. Furthermore, SIRT1 has been demonstrated to stimulate Wnt signalling by directly or indirectly deacetylating FoxOs, which hindered FoxOs interaction with β-catenin, resulting in increased β-catenin expression. The ovariectomised rat group had low SIRT1 and FoxO3a expression, indicating impaired osteoblastic development in the osteoporotic state. Furthermore, both SIRT1 and FoxO3a protein levels were higher in the MSCs and RSV combination group, showing that combination therapy was more effective [42].

FoxO1, a member of the Forkhead box O protein family, is the most common isoform in osteoblasts. By protecting against the detrimental effects of oxidative stress, FoxO1 creates an optimal intracellular environment for osteoblast proliferation and differentiation [139]. Overexpression of antioxidant enzymes such as glutathione peroxidase and superoxide dismutase by FoxO1 can inhibit ROS production [140]. FoxO1 has been found to reduce ROS in hematopoietic stem cells by raising antioxidant enzyme expression, whereas FoxO1 deletion led to an increase in bone marrow osteoclast progenitors [141]. SIRT1 appears to direct the FoxO-dependent response towards antioxidant activity and a delicate redox balance between the antioxidant and oxidant systems [142]. According to a real-time PCR study by Ameen et al. [54], FoxO1, SIRT1, and OPG gene expressions were significantly decreased in aged rats, while RANKL gene expression was increased significantly. RSV treatment increased the expressions of FoxO1, SIRT1, and OPG genes while decreasing the expression of RANKL. In an in vitro study, RSV inhibited the NFATc1 signalling pathway to prevent doxorubicin-induced osteoclast fusion and activation and to increase FoxO1 transcriptional activity [106]. Another study found that RSV increased FoxO1 transcriptional activity by inhibiting the PI3K/AKT signalling pathway, promoting oxidative stress resistance, and inhibiting osteoclastogenesis [37].

The Wnt signalling system is known to play a vital role in bone remodeling via a variety of processes, including decreasing adipogenic differentiation potential and shifting the cell fate from adipocytes to osteoblasts and chondrocytes [143]. Wnt signalling pathway also promoted osteoblast proliferation, while decreasing apoptosis [144]. The Wnt signalling pathway is classed as canonical or non-canonical based on the presence or absence of catenin [145]. The canonical Wnt/β-catenin signalling pathway, which requires β-catenin, was involved in osteoblast differentiation and bone mass retainment [146]. SIRT1 deacetylation of FoxO was discovered to limit the interaction between FoxO and β-catenin, hence boosting Wnt signalling and encouraging MSC differentiation into osteoblasts [147]. Zhao et al. [66] reported that RSV increased β-catenin expression early in MSC differentiation. SIRT1 knockdown inhibited Wnt/β-catenin signalling, whereas RSV treatment or SIRT1 overexpression activated Wnt/β-catenin signalling. This is consistent with an in vivo study revealed that Wnt/β-catenin expression was enhanced by RSV treatment in ovariectomized rats [42].

Furthermore, SIRT1 activates peroxisome proliferator-activated receptor gamma coactivator (PGC1α) transcriptional activity to induce mitochondrial biogenesis and induction of antioxidative enzymes, which can protect bone cells and inhibit mitochondrial ROS generation [148]. Ma et al. discovered that osteoporosis caused by osteomyelitis was associated with an increase in ROS in mitochondria induced by LPS, which contributed to mitochondrial dysfunction and inhibited osteogenic differentiation in MC3T3-E1 cells. SIRT1 could activate PGC-1, reduced ROS in mitochondria, and increased mitochondria biogenesis, thereby inhibiting osteogenic differentiation. In this research, Ma et al. [87] found that RSV improved LPS-inhibited osteoblast differentiation in MC3T3-E1 cells by acting on SIRT1 and PCG-1a, both of which are important regulators of mitochondrial function.

On the other hand, SIRT1 activation inhibits H_2_O_2_-induced osteoblast apoptosis via the FoxO1/β-catenin pathway [149]. SIRT1 has also been shown to suppress osteoblast apoptosis by inhibiting both the p53-p21 and the NF-κB signalling pathways [150]. Further to that, SIRT1 inhibited osteoclast differentiation by negatively regulating NF-κB and positively regulating FoxO transcription factors [151]. SIRT1-associated enzymes are histone acetylation enzymes, which can regulate NF-κB to regulate inflammation and stimulate osteoclastogenesis. According to Feng et al. [39], RSV induced SIRT1 activation and subsequent NF-κB suppression in ovariectomised rats. Likewise, RSV-mediated SIRT1 activation has been shown to inhibit the NF-kB signalling pathway, promoting osteoblast differentiation [104]. Figure 2 depicts the mechanisms underlying RSV’s action in promoting osteoblast proliferation and differentiation by modulating the SIRT1 signalling pathway.

### 3.3. Regulating the MAPKs/JNK/ERK and PI3K/AKT Signaling 

Mitogen-activated protein kinases (MAPKs) are dual (Tyr and Ser/Thr) protein kinases, which include p38 protein kinases, c-Jun N-terminal kinases/stress-activated protein kinases (JNKs/SAPKs), and extracellular signal-regulated kinase 1/2 (ERK1/2) or p44/42 MAPK [152]. MAPKs have critical roles in signal transduction as well as various biological functions such as cell viability and division, survival, and apoptosis [153,154]. Because MAPK-mediated signalling pathways have been shown to protect cells against apoptosis, they must play a function in cell proliferation and differentiation [154]. Previously, ERK1/2 was connected to cell proliferation, whereas p38 was linked to cell differentiation [155]. JNK activity is required for the initiation of early osteogenic differentiation in MSCs [156,157]. 

Phosphatidylinositol 3-kinase (PI3K) is a type of phosphate kinase that has been linked to cell signalling pathways that affect cellular death and longevity [158]. Activation of PI3K receptors causes phosphorylation of phosphatidylinositol to form phosphatidylinositol 3,4,5-triphosphate as the second messenger, with phosphatidylinositol 4,5-bisphosphate as the substrate. This causes activation of downstream signalling molecules such as AKT [159]. Both RANKL and MCSF activated the PI3K/AKT signalling pathway, which has been shown to regulate osteoclast survival and differentiation [160]. Cyclic-AMP (cAMP)-response element binding protein (CREB) is known to be involved in the transcriptional regulation of inflammatory mediators [161]. On the other hand, M-CSF-activated MAPK signalling is primarily involved in the regulation of osteoclast precursor proliferation which activates inflammatory mediators such as iNOS, COX-2, TNF- α, and several interleukins [162]. Thus, p38 MAPK signalling is critical for regulating cellular processes, particularly inflammation.

RSV has been shown to activate ERK1, ERK2, and p38, as well as to promote the proliferation and maturation of mouse osteoblasts [65]. Furthermore, RSV treatment of canine BMSCs significantly increased osteogenic markers ALP activity, and calcium nodules by activating the Wnt and MAPK/ERK signalling pathways (Figure 3) [66]. Concomitantly, Mei et al. [74] found that RSV pre-treatment provided significant protection against cadmium-induced apoptosis and attenuated cadmium-induced inhibition of osteogenic differentiation by modulating MAPK/ERK1/2/JNK signalling. Meanwhile, another study found that RSV was able to reduce immune cell activation and the subsequent synthesis and release of proinflammatory mediators by inhibiting CREB and MAPK signalling pathways in RAW264.7 cells [105]. Further to that, RSV inhibited the PI3K/AKT signalling pathway and increased FOXO1 transcriptional activity as well as endogenous antioxidants such as SOD and glutathione peroxidase, thereby inhibiting osteoclast proliferation and differentiation [37].

### 3.4. Regulating microRNA and BMP2 Signalling

MicroRNAs (miRNAs) are single-stranded, noncoding RNAs (18–25 bases) that primarily inhibit their target genes at the post-transcriptional level [163]. MiRNAs can inhibit post-transcriptional gene expression by interfering with target messenger RNA (mRNA) translation, and their functions in numerous cellular and molecular processes, including MSC osteogenic development, have been widely explored [164,165]. Furthermore, MiRNAs control osteogenic differentiation by targeting transcription factors (TFs) and genes that encode either negative or positive differentiative modulators [166]. For instance, when MSCs were stimulated to differentiate into osteoblasts, the levels of miR-27a, miR-489, and miR-148b were reduced [167]. As a result, miRNAs play a significant role in the differentiation of MSCs into osteoblasts.

The bone synthesis signalling pathways BMP-2/Smad/Runx2 are associated to osteoblast differentiation and extracellular matrix synthesis [168,169]. BMP2 is a multifunctional protein that interacts with receptors on the target cell membrane. The BMP-2 ligand secreted by osteocytes bound to cell membrane receptors, BMPRI and BMPRII, forming a complex. Type II receptor phosphorylation activated type I receptor, which then activated and phosphorylated receptor regulated Smads protein. Regulated Smads bind to Smad 4 and translocate to the nucleus. The complex binds to DNA binding protein and regulates the expression of downstream transcription factors RUNX2 and osterix (OSX), thereby regulating the transcription of genes associated with various stages of osteogenic differentiation [170]. In this pathway, Borsani et al. [91] demonstrated that RSV improved osteoblast proliferation, differentiation, and mineralisation in bisphosphonate-treated osteoblasts via the BMP2 and SIRT1 signalling pathways.

NADPH Oxidase 4 (Nox4) is primarily secreted by osteoclasts, and its activation promotes osteoclast differentiation and maturation, promoting bone resorption and, ultimately, osteoporosis. A study on Nox4 knockout mice revealed increased bone density as well as a decrease in the number of osteoclasts and their markers [171]. In C57BL/6 J mice, a high-fat diet can result in increased Nox4 expression, which leads to increased NF-κB-p65 expression [172]. Furthermore, Nox4 is a miR-128 miR-92b-3p regulatory target that regulates BMSC proliferation and differentiation [45]. According to Zhang et al. [45], the activity of BMP-2/Smad/Runx2 signalling pathway was reduced in OVX rats which led to osteoporosis. Suppression of miR-92b-3p expression increased Nox4 and NF-κB levels, decreased Smad7, BMP2, and RUNX-2 genes and protein expression, and inhibited BMSC proliferation and osteoblast differentiation, resulting in worsened osteoporosis. In contrast, when OVX rats were treated with RSV, the Nox4/NF-κB pathway activity was inhibited by the high miR-128 and miR-92b-3p levels, as well as the BMP-2/Smad/Runx2 signalling pathway activity, which could alleviate postmenopausal osteoporosis (Figure 4). Another intriguing study from Guo et al. [46] found that RSV suppressed miR338-3p, followed by an increase in RUNX2 expression in OVX rats.

### 3.5. Induction of Osteoclastogenesis by RANKL-RANK, NF-κB and NFATc1 Signaling

RANKL plays a critical role in the formation, differentiation, and survival of osteoclasts. RANKL binds to RANK which subsequently stimulates the key signalling pathways such as NF-κB, nuclear factor of activated T cells 1 (NFACTc1), AKT, p38, JNK, and ERK, all of which are important for osteoclast development and survival [173,174]. NFATc1 is a master regulator of osteoclastogenesis that is controlled by the transcription factors NF-κB and c-Fos [175]. Physiologically, NF-κB is mostly found in the cytosol as an inactive dimer, bound to its partner, and inhibitor of NF-κB (iκB). RANKL therapy can cause IκB disintegration, allowing the NF-κB unit to translocate into the nucleus and initiate transcription, as well as being involved in the activation of NFATc1. NFATc1 is activated in osteoclast progenitors by c-Fos and ERK activation [176]. NFACTc1 activation then promotes the production of osteoclast-specific genes such as Acp5 (encoding tartrate-resistant acid phosphatase [TRAcP]), Ctsk (encoding cathepsin K), and Mmp9 (encoding matrix metalloproteinase 9), resulting in osteoclastogenesis [177,178]. As a result, inhibiting the development of osteoclasts by targeting these signalling pathways is a promising route for osteoporosis treatment.

The potential of RSV’s anti-bone resorbing effects on the murine macrophage cell line RAW264.7 cells was identified through in vitro studies. Intracellular reactive oxygen species (ROS) serve as significant signalling molecules which regulate osteoclastogenesis. Previous research has shown that ROS increases osteoclast development. RANKL-stimulated osteoclast progenitors generate significantly more ROS, increasing osteoclastogenesis and bone resorption [179]. Increased intracellular ROS increased IκB phosphorylation and degradation by IκB kinase (IKK), allowing NF-κB dimers to translocate to the nucleus [180]. Previous research found that inhibiting NFATc1 in RAW 264.7 cells entirely eliminated RANK-induced osteoclastogenesis [181,182]. In line with the in vitro results, RSV therapy inhibited RANKL, demonstrating RSV’s ability to inhibit ROS and NFATc1-mediated osteoclastogenesis [37,106].

## 4. Bioavailability and Clinical Perspective of RSV

RSV is well absorbed orally but has poor bioavailability in rodents and humans. It is rapidly metabolised and has a low bioavailability, whereas trace levels of unmodified RSV are present in the systemic circulation and have less side effects [183]. In humans, roughly 70% of RSV (25 mg) is rapidly absorbed by the gastrointestinal system and metabolised (in 30 min), resulting in a peak plasma level of 2 microM of RSV and its metabolites (extremely low bioavailability) and a half life of 9 to 10 h [184]. RSV is conjugated by uridine-diphospho-glucoronosyl-transferases (UGTs) in the liver to produce two metabolites, trans-RSV-4′-O-glucuronide (G1) and trans-RSV-3-O-glucuronide (G2) [183,184].

RSV is safe and not carcinogenic, does not cause acute skin and eye irritation, or other allergic symptoms. Trans-RSV is also well tolerated by humans, with 450 mg of RSV per day being a safe dose for a 70 kg person [185]. A 2019 meta-analysis of six randomised controlled trials on bone biomarkers discovered that RSV significantly increased serum and bone ALP levels (biomarkers for bone formation) [186]. Another study in Type 2 diabetes mellitus patients found that taking 500 mg RSV daily for six months, improved whole-body BMD, bone mineral content, and T-score (a measure of bone density relative to a 30-year-old person) [187]. Furthermore, a 12-month supplementation of RSV at the dose of 150 mg per day to postmenopausal women was found to significantly improve the BMD of the lumbar spine and neck of the femur, as well as a reduction in C-terminal telopeptide type-1 collagen levels (a bone resorption marker), but not for the whole-body BMD [188]. In both studies, improvements were greater in patients with poor baseline bone parameters.

## 5. Conclusions

The purpose of this article was to review in vitro and in vivo studies of RSV to highlight the compound’s osteogenic and osteoinductive properties. Preclinical evidence lays the groundwork for future human studies to investigate the potential benefit of RSV supplementation, which is crucial for lowering the population’s risk of osteoporosis. Collectively, RSV’s capacity to function as both an anabolic and antiresorptive agent makes it an attractive candidate medication for the prevention and treatment of osteoporosis. The in vivo studies are important in osteoporosis research because they provide documentation of RSV’s beneficial effects, notably on bone-forming action, which may transfer into therapeutic uses in clinical practices and dietary advice to reduce bone loss. RSV has been shown in studies utilising postmenopausal, senile, and disuse osteoporosis models to have bone-protective properties. However, variations in food intake, body weight, and endocrine function in the osteoporosis models may have confounded some of the RSV’s bone effects. There is also no consensus on the appropriate method for RSV dose translation from animal species to humans. Apart from that, RSV’s bone-protective effects should have been compared to the standard anti-osteoporotic treatments.

In the in vitro study, RSV was thought to be a SIRT1 activator or directly modulate Runx2 and OSX expressions to regulate osteoblast differentiation and RANKL/OPG, NFACT1 and NFκB expressions to regulate osteoclast differentiation. These dual actions of RSV on osteoblast and osteoclast are beneficial in maintaining balance in bone remodelling, a key endpoint in the management of osteoporosis. Furthermore, RSV enhanced MSC growth, proliferation, and differentiation to osteoblast, which may be advantageous for applications in stem cells or scaffolds to improve bone restoration.

## Figures and Tables

**Figure 1 biomedicines-11-01453-f001:**
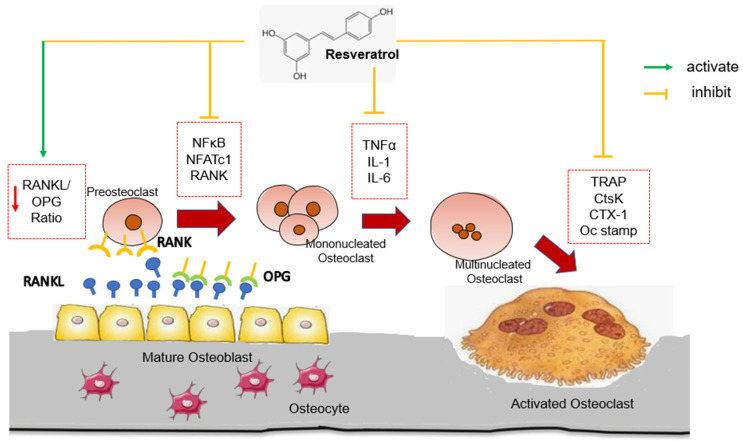
The role of resveratrol (RSV) in regulating osteoclastogenesis through the RANK/RANKL/OPG system. RSV alters the expressions of OPG and RANKL by osteoblasts. RSV inhibits the maintenance of osteoclast lineage commitment, osteoclast maturation, and bone resorption in osteoclasts by inhibiting NF-κB and NFATc1, and downregulating the proinflammatory cytokines such as TNFα, IL-1 and IL-6. Abbreviation: OPG: osteoprotegerin; RANK: receptor activator of nuclear factor-kappa B; RANKL: receptor activator of nuclear factor-kappa B ligand; NF-κB: nuclear factor-kappa B; NFATc1: nuclear factor of activated T-cells cytoplasmic 1; TNFα: tumor necrosis factor α; IL-1: interleukin-1; IL-6: Interleukin-6; CtsK: cathepsin K; Oc-Stamp: osteoclast stimulatory transmembrane protein; TRAP: tartrate-resistant acid phosphatase. CTX-1: C-terminal telopeptides type I collagen.

**Figure 2 biomedicines-11-01453-f002:**
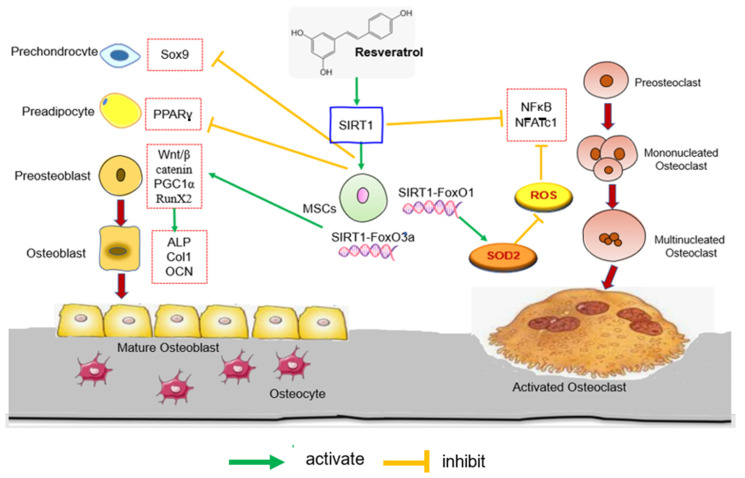
The effects of RSV on activation of SIRT1 signalling pathway. The activation of the SIRT1-FoxO3a complex transcription in MSCs promote osteoblast differentiation while suppressing PPARγ and Sox9 to inhibit adipocyte and chondrocyte differentiation, respectively. Meanwhile, the transcriptional activity of SIRT1-FoxO increases SOD2 production to reduce ROS production and inhibits the NF-κB and signalling pathway, which favour osteoclastogenesis. Abbreviation: SIRT1: silent information regulator of transcription 1; FoxO: Forkhead box O; RunX2: runt-related transcription factor; PGC1α: peroxisome proliferator activated receptor gamma coactivator; ROS: Reactive Oxygen Species; SOD: Superoxide dismutase; NFATc1: nuclear factor-activated T cells; PPARγ: Peroxisome proliferator-activated receptors γ; NF-κB: Nuclear factor-κB; ALP: Alkaline phosphatase; Col1: Collagen type 1; OCN; Osteocalcin.

**Figure 3 biomedicines-11-01453-f003:**
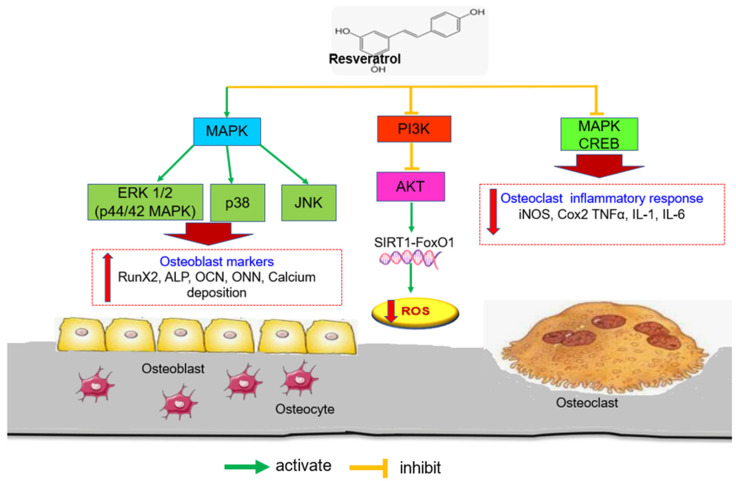
The effects of RSV on MAPK (ERK 1/2/, p38 MAPK and JNK) signalling pathways. The activation of MAPKs pathway results in the upregulation of osteogenic genes which favours osteoblastogenesis. FoxO1 transcriptional activity increases in response to PI3K/AKT signalling inhibition, which lowers oxidative stress. The generation of inflammatory cytokines that promote osteoclast differentiation or activation can be reduced through CREB phosphorylation in MAPK signalling suppression. Abbreviations: MAPK: Mitogen-activated protein kinases; ERK 1/2: extracellular signal-regulated kinase 1/2; JNK: c-Jun N-terminal kinases/stress-activated protein kinases; RunX2: runt-related transcription factor; ALP: Alkaline phosphatase; OCN: Osteocalcin; ONN: Osteonectin; PI3K: Phosphatidylinositol 3-kinase; Akt: protein kinase B; CREB: Cyclic-AMP (cAMP)-response element binding protein; iNOS: nitric oxide synthase; Cox2: cyclooxygenase-2; TNFα: tumour necrosis alpha; IL-1: Interleukin-1; IL-6: Interleukin-6.

**Figure 4 biomedicines-11-01453-f004:**
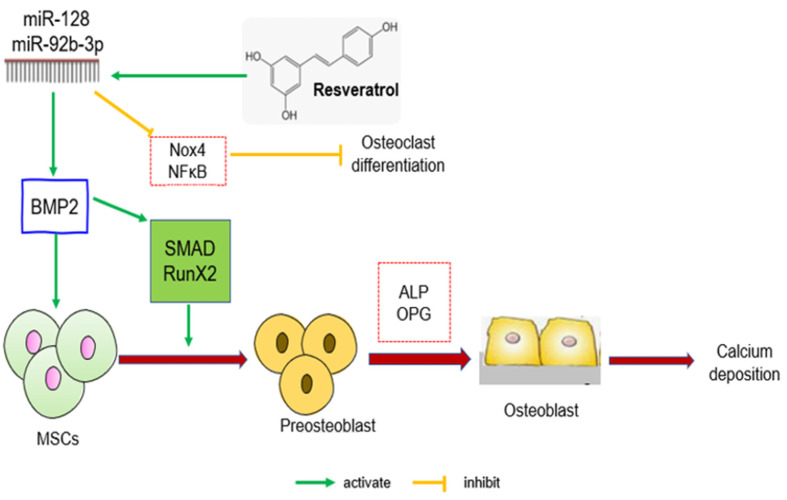
The effects of RSV on the regulation of microRNAs and BMP2 signalling in protecting bone. The interaction of specific microRNAs and BMP2 results in the downregulation of Nox4/NF-B pathway activity and upregulation of SMAD/RUNX2 protein to promote osteoblast proliferation, differentiation, and calcium deposition. Abbreviation: miR: microRNA; BMP2: bone morphogenetic protein 2; SMAD: Suppressor of mothers against decapentaplegic. Nox4: NADPH Oxidase 4; NF-κB: Nuclear factor-κB; ALP: Alkaline phosphatase; RUNX2: runt-related transcription factor; OPG: osteoprotegerin.

**Table 1 biomedicines-11-01453-t001:** The effects of RSV on bone in in vivo studies. Symbol ↑ indicates an increase, ↓ indicates a decrease, and ↔ indicates no change.

Type of Animal and Induction	Intervention (Dose, Route, and Duration)	Research Findings	Mode of Action	References
Post menopausal osteoporosis				
Adult unmated female Sprague Dawley (SD) rats, 3-month-old, and bilateral ovariectomy	RSV (5, 15, and 45 mg/kg/day), oral, and 12 weeks	Serum estrogen ↑, serum Ca ↔, Serum P ↔, TRAP ↓, PINP ↑, ALP ↔, CTX-1 ↓, DPD ↔, and BMD ↑	RSV increased bone mineral density by downregulating osteoclastogenesis	Zhang et al., 2020 [33]
Female Wistar Rats 3 to 4-month-old, bilateral ovariectomy	RSV (20, 40, 80 mg/kg/day), oral, 8 weeks	BMD ↑, Tb Ar ↑, Tb Th ↑, Tb sep ↓, Tb N ↑, Serum Ca ↔, Serum P ↔, ALP ↑, TRAP ↓, IL-6 ↓, TNFα ↓, OPG ↑, RANKL ↓	RSV promoted osteogenesis by downregulating pro-inflammatory pathway and upregulating OPG/RANKL ratio	Zhao et al., 2014 [35]
Female Sprague Dawley (SD) rats, 3-month-old, and bilateral ovariectomy	RSV (625 µg/kg/day), feeding diet pellets, 4 weeks	BMD ↑, RANKL/OPG ↓, IL-23 ↓, IL-17a ↓, IL1β ↓, TNFα↓	RSV promoted osteogenesis by downregulating pro-inflammatory pathway and RANKL/OPG ratio	Khera et al., 2019 [36]
Female Sprague Dawley (SD) rats, 3-month-old, bilateral ovariectomy	RSV (40 mg/kg/day), subcutaneous injection, and 10 weeks	BMD ↑ (right femur), BV/TV ↑, Tb.Th ↑, Tb.N ↑, Ct.Vol↑, Ct.Th ↑, Conn D ↑, TRAP↓, RANKL/OPG ↓, MDA ↓, ROS ↓, GSH ↑, and SOD ↑	RSV attenuated oxidative stress damage and RSV-suppressed osteoclastogenesis	Feng et al., 2018 [37]
Female mice, 8-week-old, and bilateral ovariectomy	RSV (40 mg/kg/day), intraperitoneal injection, and 8 weeks	BV/TV ↑, Tb Th ↑, Tb sep ↓, ALP ↑, RunX2 ↑, Ob Nu ↑, Oc Nu ↓, TRAP ↓ Osx ↑, P1NP ↑, Sirt1 ↑, and SOD ↑	RSV induced osteogenesis via activation of the SIRT1/FoxO1 signalling pathway.	Jiang et al. 2020 [38]
Female Sprague Dawley (SD) rats, 3-month-old, and bilateral ovariectomy	RSV (5, 25, 45 mg/kg/day), oral, and 8 weeks	BMD ↑, BV/TV ↑, Tb Th ↑, Tb sep ↓, ALP ↑, OPN ↑, Col1a1↑, Sirt1 ↑, and NF-κB ↓	RSV increased osteoblast differentiation by activating SIRT1 and inhibiting the NF-kB signalling pathway.	Feng et al., 2014 [39]
Female albino rats, 3-month-old, and bilateral ovariectomy	RSV (80 mg/kg/day), oral, and 8 weeks	BMC ↑, BMD ↑, PINP ↓, ALP ↓, CTX-1 ↓, RANKL/OPG ↓, Sirt1 ↑, FOXO3a ↑, Wnt ↑, β catenin ↑, and RunX2 ↑	RSV-promoted osteogenesis through Sirt1 and Wnt signalling pathways.	Elseweidy et al., 2021 [42]
Sprague Dawley (SD) rats, 6-month-old, and bilateral ovariectomy	RSV (50, 100, 200 mg/kg/day), oral, and 12 weeks	BMD ↑, Tb Ar ↑, Tb Th ↑, Tb sep ↓, Tb N ↑, miR-92b-3p ↑, NF-κB ↓, Nox4 ↓, CtsK ↓, Smad7 ↑, BMP2 ↑, RUNX-2 ↑, ALP ↑, OPG ↑, TRAP ↓, CTX-1 ↓, Ca ↑, and P ↑	RSV increased miR-92b-3p transcriptional activity to enhance the BMP-2/Smad/Runx2 signalling and reduced NF-κB to promote osteogenesis.	Zhang et al., 2020 [45]
Female Wistar rats, 8 –week old, and bilateral ovariectomy	RSV (500 mg/kg/day), tail intravenous injection, and 1 day	BMD ↑, ALP ↑, Ca ↑, miR3383p ↓, RunX2 ↑	RSV decreased miR 338 3p expression, which was followed by a rise in RunX2 expression.	Guo et al., 2015 [46]
Female Sprague Dawley (SD) rats, 10 to 12-week-old, bilateral ovariectomy	RSV (10, 20, and 40 mg/kg/day), oral, 8 weeks	BMD ↑ (lumbar L3, femur-tibia), Ca content ↑, VEGF ↑, Col1a1 ↑, RANKL ↓, autophagy related genes, atg 5, atg 7, and atg 12 ↑	RSV promoted osteoblastic differentiation and suppressed osteoclastic differentiation by regulating autophagy	Wang et al., 2020 [50]
Senile osteoporosis				
Aged male Wistar rats and 22 –month old	RSV (10 mg/kg/day), oral, and 10 weeks	BV/TV ↑, Tb Th ↑, Tb sep ↓, Cr Th ↑, CTX ↔, and OCN ↔		Tresguerres et al., 2014 [53]
Aged male Wistar albino rats and 18 to 20 –week old	RSV (20 mg/kg/day), oral, and 6 weeks	BMD ↔, OPG ↑, RANKL ↓, FoxO1 ↑, Sirt1 ↑, GSH ↑, MDA ↓, hsCRP ↓, TNFα↓, IL6 ↓, and IL1β ↓	RSV showed anti-osteoporotic effects with involvement of FoxO1/SIRT/RANK/OPG pathways	Ameen et al., 2020 [54]
Aged male Wistar Rats and 6 –month old	RSV (20 mg/kg/day), oral, and 3 months	BV/TV ↔, cortical bone volume ↔, Sirt1 ↔, Osx ↔, OCN ↑, CTX-1 ↓, and ALP ↔,		Lee et al., 2014 [55]
Senile and diuse osteoporosis				
Aged male Fischer 344 9 Brown Norway, 33-month-old, and hindlimbs suspended for 14 days	RSV (12.5 mg/kg/day), oral, and 3 weeks	BV/TV ↔, Tb Th ↑, Tb sep ↓, ALP ↑, OCN ↑, C-reactive protein ↓, TNFα↓, Ca ↔, P ↔, TRAP ↔, and CTX1 ↔	RSV increased osteoblast bone formation and possibly due to reduced inflammation.	Durbin et al., 2014 [58]
Secondary osteoporosis				
Male Sprague Dawley (SD) rats, 3-month-old, intramuscular injection of dexamethasone 5 mg/kg, and twice a week 6 weeks	RSV (5, 45 mg/kg/day), oral, and 8 weeks	BMD ↑ (right femur), femoral porosity ↓, ALP ↓, OCN ↓, Sirt1 ↑, LC3 ↑, Beclin-1 ↑, phospho-mTOR ↓, and phospho-Akt ↓	RSVenhanced SIRT1 expression and protect dexamethasone-treated osteoblasts by autophagy via Akt/mTOR pathway	Yang 2019 [60]
Male Sprague Dawley (SD) rats, 6 week old, and subcutaneous injection of methotrexate for five days (once daily) 0.75 mg/kg,	RSV (1 mg/kg/day), oral, and 12 days (7 days pretreatment, and 5 days during methotrexate injection)	growth plate thickness ↑, primary spongiosa bone, BV/TV ↑, Tb Th ↑, adipose density ↓, RunX2 ↔, Osx ↓, Ocn ↔, TRAP ↓, TNF-α ↓, IL-1 ↓, and IL-6 ↓	RSV improved bone microstructure by inhibiting osteoclastogenesis and inflammatory processes.	Lee 2017 [62]

Abbreviation: BMD: Bone mineral density; BV/TV: bone volume/Total volume; Tb.Ar: Trabecular Area; Tb.Th: Trabecular thickness; Tb.N: Trabecular Number; Tb.Sp: Trabecular space, Conn D; Connective Density; Ct. Vol: Cortical Volume; Ct. Th: Cortical Thickness; ALP: Alkaline phosphatase; OCN: Osteocalcin; SIRT1: silent information regulator of transcription 1; Akt: Ak strain transforming; mTOR: *mammalian target of rapamycin*; NF-κB: Nuclear factor kappa-light-chain-enhancer of activated B cells; *VEGF*: vascular endothelial growth factor; Col1a1: alpha-1 type I collagen; TNFα: tumour necrosis factor alpha; MDA: *Malondialdehyde*; *SOD*: Superoxide dismutase; GSH: Gluthathione; FoxO1: Forkhead box O1; PI3K/Akt: phosphatidylinositol-3-kinase; RANKL: receptor activator of nuclear factor-κB ligand; OPG: osteoproprotegrin; TRAP: tartrate-resistant acid phosphatase; IL-6: Interleukin-6; IL1β: Interleukin-1beta; hsCRP: high sensitivity C reactive protein; CTX-1: C-terminal telopeptides type I collagen; BMP2: bone morphogenic protein 2; Osx: Osterix; P1NP: amino-terminal propeptide of type 1 procollagen; RunX2: runt-related transcription factor 2; Ob Nu: Osteoblast number; Oc Nu: Osteoclast.

**Table 2 biomedicines-11-01453-t002:** The effects of RSV on bone cells in in vitro studies. Symbol ↑ indicates an increase, ↓ indicates a decrease, and ↔ indicates no change.

Type of Cell and Induction	Intervention	Research Findings	Mode of Action	References
Bone forming cells				
3-D osteogenic differentiation on collagen scaffolds of rat adipose stem cells	25 µM RSV	ALP ↑, OCN ↑, OPG ↑, and Mineral density ↑		Dosier et al., 2012 [64]
HBMSC cells	10−6 M RSV	Cell proliferation ↑, Calcium deposition ↑, ALP ↑, Runx2 ↑, Osterix ↑, OCN ↑, ERK 1/2↑, ER ↑, and MAPK ↓	RSV induced HBMSC proliferation and differentiation through an ER-dependent pathway linked to ERK1/2 activation.	Dai et al., 2007 [65]
BMSC cells	20 µM RSV	Cell proliferation ↑, Calcium deposition ↑, MMP ↔, ALP ↑, Runx2 ↑, ONN ↑, OCN ↑, GSK-3b ↑, β catenin ↑, ERK ↑, and MAPK ↑	RSV promoted osteoblastic differentiation of BMSCs by activating the Wnt/β-catenin and ERK/MAPK signalling pathways.	Zhao et al., 2018 [66]
Senescent BMSC cells	5,10, 15, 20, and 25 µM RSV	Cell viability ↑, ALP ↑, Col-I ↑, OCN ↑, OPN ↑, RunX2 ↑, ROS ↓, p16 ↓, p21 ↓, p53 ↓, and AMPK ↑	RSV promoted osteogenic differentiation of senescent BMSCs by slowing cell ageing and decreasing ROS production via AMPK activation.	Zhou et al., 2019 [70]
HBMSC cells, hypoxia	0.1 and 1 µM RSV	Cell viability ↑, OCN ↑, OPN ↑, ALP ↑, RunX2 ↑, and ROS ↓	RSV reduced high-altitude hypoxia-induced osteoporosis by increasing osteoblastogenesis and inhibiting the ROS/HIF-1 signalling pathway.	Yan et al., 2022 [72]
MC3T3-E1 cells, induced with 120 µM H_2_O_2_	15 µM RSV	p53 ↓, Bax ↓, Bcl-2 ↓, and caspase 9 ↓	RSV suppressed oxidative stress-induced apoptosis in osteoblasts.	He et al., 2015 [73]
MC3T3-E1 cells, induced with 5 µM cadmium	10 µM RSV	Cell viability ↑, ALP ↑, Col 1 ↑, BMP2 ↑, RunX2 ↑, ERK ↑, and JNK ↑	RSV protected MC3T3-E1 cells from cadmium-induced osteogenic differentiation suppression by modulating the ERK1/2 and JNK pathways.	Mei et al., 2021 [74]
human periosteum derived MSCs (PO-MScs)	5 µM RSV	Cell proliferation ↑, Calcium deposition ↑, ALP ↑, mitochondrial mass ↑, and mitochondrial DNA copy number ↑	RSV stimulated mitochondrial biogenesis in the process of PO-MSC osteogenic differentiation.	Moon et al., 2020 [79]
MC3T3-E1 cells,0.5, 1, and 2 μg/mL Lipopolysaccharides(LPS)-Induced Inhibition of Osteoblast	25 µM RSV	Cell viability ↑, ALP ↑, OCN ↑, OPN ↑, RUNX2 ↑, Sirt1 ↑, and PGC-1α ↑	RSV reduced the inhibition of LPS on osteoblast development via enhancing mitochondrial activity and SIRT1 pathway	Ma et al., 2022 [87]
HBMSC cells, 1 μg/mL Lipopolysaccharides	25 µM RSV	Cell proliferation ↑, ALP ↑, PINP ↑, OPG ↑, LDH ↔, IL-6 ↔, and IL-8 ↔	RSV promoted osteoblast differentiation independently of inflammation.	Ornstrup et al., 2015 [88]
HOB cells, 5 µM alendronate and zoledronate	10 µM RSV	Cell viability ↑, BMP2↑, OPG ↑, Col1 ↑, Sirt1 ↑, and Calcium deposition (Alizarin Red) ↑	RSV improved the osteoblast proliferation, differentiation and mineralization that treated with alendronate and zoledronate, through BMP2, and SIRT1 signalling pathway	Borsani et al., 2018 [91]
MC3T3-E1 cells, induced with 10 µM PGE2	50 µM RSV	OPG ↓, p44/p42 MAP kinase ↓, p38 MAP kinase ↓, SAPK/JNK ↓, and Sirt1 ↔	Resveratrol inhibited PGE2-stimulated OPG synthesis in osteoblasts by inhibiting p44/p42 MAP kinase, p38 MAP kinase, and SAPK/JNK, and these suppressive effects are independent of SIRT1 activation.	Yamamoto et al., 2014 [98]
MC3T3-E1 cells, induced with 10 µM PGF2α	10 µM RSV	OPG ↓, p44/p42 MAP kinase ↓, p38 MAP kinase, SAPK/JNK↓, and SIRT1 ↑	RSV suppressed PGF2-stimulated OPG synthesis in osteoblasts via the MAP kinase pathways and these suppressions are mediated by SIRT1	Kuroyanagi et al., 2014 [99]
Primary rat calvarial osteoblasts	0.05, 0.1 μM RSV	Cell viability ↑, Calcium deposition ↑, ALP ↑, ERα ↑, and ERβ ↔	RSV stimulated osteoblast differentiation via an oestrogen-dependent pathway	Shah et al., 2022 [102]
human embryonic MSCs	5 µM RSV	RunX2 ↑, ALP ↑, Ocn ↑, SIRT1 ↑, FoxO3a ↑, and SIRT1-FoxO3a binds to FoxO response element (FRE) site	RSV promoted osteogenesis of human mesenchymal stem cells by upregulating RUNX2 gene expression via the SIRT1/FoxO3a axis.	Tseng et al., 2011 [103]
Canin MSCs	5 µM RSV	Col1 ↑, Ocn ↑, β1- Integrin ↑, RunX2 ↑, SIRT1 ↑, TNFβ ↓, and NF-κB ↓	RSV suppressed TNFβ expression by activation of SIRT1 and inhibition of NFκB signalling pathway.	Costanze et al., 2020 [104]
Bone resorbing cells				
RAW 264.7 cells, induced with 5 µg/mL LPS	1, 5, 10 µM RSV	Cytotoxic effect ↔, Nitric oxide ↓, PGE2 ↓, iNOS ↓, COX-2 ↓, TNF-α ↓, IL-1β ↓, CREB ↓, MAPK ↓, PI3K/AKT ↓, and Sirt1 ↑	RSV inhibited proinflammatory mediators and cytokines production in response to LPS by activation SIRT1 expression, and inhibition of PI3K/AKT, and CREB and MAPK signalling pathway	Zong et al., 2012 [105]
RAW 264.7 cells, induced with 10-4M H2O2	10-5M RSV	Cytotoxic effect ↔, MMP-9 ↓, TRAP ↓, CtsK ↓, MDA ↓, ROS ↓, SOD ↑ GSH-PX ↑, FoxO1 ↑, and PI3K/AKT ↓	RSV increased FoxO1 transcriptional activity by inhibiting the PI3K/AKT signalling pathway, promoting oxidative stress resistance, and inhibiting osteoclastogenesis.	Feng et al., 2018 [37]
RAW 264.7 cells, 0.1 µM doxorubicin-Induced Osteoclast Differentiation	10 µM RSV	Cytotoxic effect ↔, Oc-Stamp ↓, RANK ↓, TRAP ↓, CtsK ↓ NFATc1 ↓, FoxO1 ↑, SOD 1 ↑, and Nrf 2 ↑	RSV prevented doxorubicin-induced osteoclast fusion and activation and increased FoxO1 transcriptional activity by inhibiting the NFATC1 signalling pathway	Poudel et al., 2022 [106]

Abbreviation: BMSC: bone mesenchymal stem cells; HBMSC: Human bone mesenchymal stem cells; FoxM1; Forkhead Box M1; Oc-Stamp; osteoclast stimulatory transmembrane protein; RANK: Receptor activator of nuclear factor κ B. TRAP: Tartrate-resistant acid phosphatase; CtsK; Cathepsin K; NFATc1; nuclear factor-activated T cells; Nrf2: nuclear factor E2-related factor 2; PGE2: prostaglandin E_2_; FoxO1: Forkhead box O1; PIK/AKT: phosphatidylinositol-3-kinase; MDA: *Malondialdehyde*; *SOD*; Superoxide dismutase; GSH: Glutathione; ROS: Reactive Oxygen Species; MAP kinase: mitogen-activated protein kinase; SAPK/JNK: stress-activated protein kinase/c-*Jun* N-terminal kinase; NFATc1: nuclear factor of activated T cells; CaN: Calcineurin; AMPK: AMP-activated protein kinase; ONN: Osteonectin; OCN: Osteocalcin; OPN: Osteonectin; MMP: matrix metalloproteinase; GSK-3b: Glycogen synthase kinase-3 beta; ERK: extracellular signal-regulated kinase; CREB: cyclic AMP-responsive element-binding protein; LPS: Lipopolysaccharide; NO: nitric oxide; iNOS: inducible nitric oxide synthase; COX-2: cyclooxygenase-2; TNF-α: tumour necrosis factor-α; IL-1 β: interleukin-1β; IL-6: Interleukin-6; LDH: Lactate Dehydrogenase; PINP: Procollagen type I N-terminal propeptide.

## Data Availability

Not applicable.

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
