# Peer review of "Revisiting Resveratrol as an Osteoprotective Agent: Molecular Evidence from In Vivo and In Vitro Studies"

_biomedicines, 2023, doi:10.3390/biomedicines11051453_

Round 1

Reviewer 1 Report

The review is well written. about RSV.

I should rearrange the column of authors reference in the tables.

Namely, it is easy understanding that they should be last (right) column. 

Author Response

Reviewer 1

The review is well written. about RSV.

I should rearrange the column of authors reference in the tables.

Namely, it is easy understanding that they should be last (right) column. 

Answer:

The table has been amended according to your suggestion. Thank you

Reviewer 2 Report

The paper regarding “Title: Revisiting Resveratrol as an Osteoprotective Agent: Molecular Evidence from In vivo and In vitro Studies ” is presented. The authors summarised and discussed the recent progress of the molecular evidence of Resveratrol as an Osteoprotective Agent. The study is potentially interesting.  

There are some possible issues

It was mentioned in Figure 1 that resveratrol (RSV) regulates osteoclastogenesis by inhibiting NF-κB.  Previous studies have found (for example PMID: 19046922 ) that NF-kB in osteoclast consists of both classical and alternative pathways of NF-kB are important for bone resorption and osteolysis. It would be relevant to discuss the effect of Resveratrol on classical and alternative pathways of NF-kB, and related signalling molecules TRAF6, aPKC, p62/SQSTM1, and deubiquitinating enzyme CYLD in osteoclasts, which might provide more in-depth information of NF-kB signalling pathway.

It was suggested in figure 2 that ROS production might favour osteoclastogenesis.  More recent studies have found that ROS in osteoclasts are involved in ROS/Keap1/Nrf2 signaling axis (for example, PMID: 36640671, PMID: 34492273; PMID: 34149419).  It would be relevant to include and discuss this key signaling axis of ROS affected by  Resveratrol in osteoclasts, which might provide more insightful ROS signalling.

This paper discussed mainly the role of Resveratrol osteoblastogenesis and osteoclastogenesis. Previous studies (for example PMID: 22465238) have shown that bone homeostasis is regulated by many cell types such as macrophages, osteoclasts, osteoblasts, bone lining cells, osteomacs, vascular endothelial cells, B cells, and T cells etc. It would be informative to discuss how Resveratrol might regulate cell types in the bone local environment. Considering, for instance, that RSV was used as a supplement as mentioned in this paper, which would affect many cells in the bone local environment

There are possible typos, for example,

supplements for prevention of osteoporosis?? supplements for the prevention of osteoporosis?

Glucocorticoid-induced osteoporosis is the most common causes?? Glucocorticoid-induced osteoporosis is the most common cause?

bed rest in human?? bed rest in humans.

…..

as above

Author Response

Reviewer 2

  1. It was mentioned in Figure 1 that resveratrol (RSV) regulates osteoclastogenesis by inhibiting NF-κB.  Previous studies have found (for example PMID: 19046922) that NF-kB in osteoclast consists of both classical and alternative pathways of NF-kB are important for bone resorption and osteolysis. It would be relevant to discuss the effect of Resveratrol on classical and alternative pathways of NF-kB, and related signalling molecules TRAF6, aPKC, p62/SQSTM1, and deubiquitinating enzyme CYLD in osteoclasts, which might provide more in-depth information of NF-kB signalling pathway.

Answer:

We are unable to provide additional information regarding the NF-kB signalling pathway since research on the effects of RSV on the detailed mechanism is limited.

  1. It was suggested in figure 2 that ROS production might favour osteoclastogenesis. More recent studies have found that ROS in osteoclasts are involved in ROS/Keap1/Nrf2 signaling axis (for example, PMID: 36640671, PMID: 34492273; PMID: 34149419).  It would be relevant to include and discuss this key signaling axis of ROS affected by Resveratrol in osteoclasts, which might provide more insightful ROS signalling.

Answer:

ROS has been discovered to be involved in the signalling of ROS/Keap1/Nrf2. However, research into RSV effects in osteoclasts for this pathway has been limited. As a result, we are unable to provide a more thorough explanation of the ROS signalling process.

  1. This paper discussed mainly the role of Resveratrol osteoblastogenesis and osteoclastogenesis. Previous studies (for example PMID: 22465238) have shown that bone homeostasis is regulated by many cell types such as macrophages, osteoclasts, osteoblasts, bone lining cells, osteomacs, vascular endothelial cells, B cells, and T cells etc. It would be informative to discuss how Resveratrol might regulate cell types in the bone local environment. Considering, for instance, that RSV was used as a supplement as mentioned in this paper, which would affect many cells in the bone local environment.

Answer

From the clinical data, they just determine the BMD, T-score and specific bone biomarkers such as ALP for bone production and C-terminal telopeptide type-1 collagen levels for bone resorption. We are unable to postulate any types of cells that are affected by resveratrol due to these constrained parameters.

  1. Typo/grammar error

Answer

All the typo error/grammar error are amended. Thank you

Reviewer 3 Report

This review article focuses on  the in vivo and in vitro experimental evidence of RSV efficacy in ameliorating bone loss  and enhancing bone formation.This review also includes information on RSV’s molecular mechanism of action and its therapeutic potential as an osteoprotective agent. In conclusion, 

 RSV has been shown to inhibit NF-κB and  RANKL-mediated osteoclastogenesis,  oxidative stress, and inflammation, while promoting osteogenesis and promoting differentiation of  mesenchymal stem cells to osteoblasts. Wnt/β-catenin, MAPKs/JNK/ERK, PI3K/AKT, FoxOs, microRNAs and BMP2 are among the possible kinases and proteins involved in the underlying mech anisms. RSV has also been shown to be the most potent SIRT1 activator to cause stimulatory effects  on osteoblasts and inhibitory effects on osteoclasts. RSV may thus represent a novel therapeutic  strategy for raising bone formation and preventing bone loss in the aging and postmenopausal population. 

The introduction is well written , with adequate bibliographic references . However, on page 1, line 42-3, incorrect information is provided. Although the risk of atypical fracture increases with time, an adequate benefit/risk ratio persists with its use The studies carried out in vivo have been adequately described, indicating their characteristics. The tables reflect the commented data The description of the effects on cells that influence metabolism are broadly described, specifying the possible mechanisms. The table clarify the data The mechanisms of action on the skeleton are well described, being the key graphical representation to understand the process.  

Author Response

Reviewer 3

The introduction is well written  with adequate bibliographic references . However, on page 1, line 42-3, incorrect information is provided. Although the risk of atypical fracture increases with time, an adequate benefit/risk ratio persists with its use. The studies carried out in vivo have been adequately described, indicating their characteristics. The tables reflect the commented data. The description of the effects on cells that influence metabolism are broadly described, specifying the possible mechanisms. The table clarify the data The mechanisms of action on the skeleton are well described, being the key graphical representation to understand the process. 

Answer

Line 42-43

However, most clinical trials provide scant information regarding long-term efficacy and safety. For example, observational data from the study of Black et al. [10] indicated that oral biphosphonate may cause atypical femoral fracture with prolonged use of biphosphonate. The most concerning fact is that the benefit-risk balance may become negative in the long term, particularly in patients with moderate osteoporotic fracture risk.

New statements

However, the use of these pharmacologic agents is associated with rare serious side effects. For example, long-term use of biphosphonate has been associated to an increase risk of atypical fracture and osteonecrosis of the jaw [10,11] Apart from that, patients may also experience common adverse effects which are mainly related to gastrointestinal, cardiovascular and endocrine system [12]. Though biphosphonate remain the first-line therapy for the treatment of osteoporosis, the concerns over its side effects have limit patients’ compliance and trust which contributed to the decline in the use of this class of drug [10,11].

Round 2

Reviewer 2 Report

The study is potentially interesting.  The molecular mechanisms of Resveratrol as an Osteoprotective Agent are lacking, and not fully presented.  It was suggested to insightfully discuss the possible role of Resveratrol on RANKL-mediated NF-kB signaling pathways and ROS pathways, as well as TCM-like effects.

Author Response

Thank you for suggestion. Below is the additional line for the possible role of Resveratrol on RANKL-mediated NF-kB signaling pathways and ROS pathways

3.5 Induction of osteoclastogenesis by RANKL-RANK, NF-κB and NFATc1 signaling

RANKL plays a critical role in the formation, differentiation, and survival of osteoclasts. RANKL binds to RANK, which subsequently stimulates the key signalling pathways such as NF-κB, nuclear factor of activated T cells 1 (NFACTc1), AKT, p38, JNK, and ERK, all of which are important in osteoclast development and survival [173, 174].  NFATc1 is a master regulator of osteoclastogenesis that is controlled by the transcription factors NF-κB and c-Fos [175]. NF-κB is mostly found in the cytosol as a dimer composed of iκB and NF-κB, and iκB can inhibit NF-κB from entering the nucleus. RANKL therapy can cause IκB disintegration, allowing the NFκB unit to translocate into the nucleus and initiate transcription, as well as being involved in the activation of NFATc1. NFATc1 is activated in osteoclast progenitors by c-Fos and ERK activation [176]. NFACTc1 activation then promotes the production of osteoclast-specific genes such Acp5 (encoding tartrate-resistant acid phosphatase [TRAcP]), Ctsk (encoding cathepsin K), and Mmp9 (encoding matrix metalloproteinase 9), resulting in osteoclastogenesis [177, 178]. As a result, inhibiting the development of osteoclasts by targeting these signalling pathways is likely a promising route for osteoporosis treatment.

Based on in vitro studies, the potential of RSV's anti-bone resorbing effects on the murine macrophage cell line RAW264.7 cells was identified. Intracellular reactive oxygen species (ROS) serve as significant signalling molecules which regulate osteoclastogenesis. Previous research has shown that ROS increases osteoclast development. RANKL-stimulated osteoclast progenitors generate significantly more ROS, increasing osteoclastogenesis and bone resorption [179]. Increased intracellular ROS increased IκB phosphorylation and degradation by IκB kinase (IKK), allowing NF-κB dimers to translocate to nucleus [180]. Previous research found that inhibiting NFATc1 in RAW 264.7 cells entirely eliminated RANK-induced osteoclastogenesis [181,182]. In line with in vitro results, RSV therapy inhibits RANKL, demonstrating RSV's ability to inhibit ROS and NFATc1 [37,106].
